# Comparison of Different Extraction Processes on the Physicochemical Properties, Nutritional Components and Antioxidant Ability of *Xanthoceras sorbifolia* Bunge Kernel Oil

**DOI:** 10.3390/molecules27134185

**Published:** 2022-06-29

**Authors:** Yuling Zheng, Pan Gao, Shu Wang, Yuling Ruan, Wu Zhong, Chuanrong Hu, Dongping He

**Affiliations:** 1Key Laboratory for Deep Processing of Major Grain and Oil of Ministry of Education in China, College of Food Science and Engineering, Wuhan Polytechnic University, 68 Xuefu Road, Wuhan 430023, China; zhongwu@whpu.edu.cn (Y.Z.); 18957650777@163.com (Y.R.); successricky@163.com (W.Z.); hcr305@163.com (C.H.); hedp123456@163.com (D.H.); 2Hubei Key Laboratory for Processing and Transformation of Agricultural Products, College of Food Science and Engineering, Wuhan Polytechnic University, 68 Xuefu Road, Wuhan 430023, China; 3Key Laboratory of Edible Oil Quality and Safety for State Market Regulation, Wuhan Institute for Food and Cosmetic Control, 1137 Jinshan Avenue, Wuhan 430012, China; wangshu8532@163.com

**Keywords:** *Xanthoceras sorbifolia* Bunge kernel oil, nervonic acid, antioxidant ability

## Abstract

In this study, we investigated and compared the oil yield, physicochemical properties, fatty acid composition, nutrient content, and antioxidant ability of *Xanthoceras sorbifolia* Bunge (*X. sorbifolia*) kernel oils obtained by cold-pressing (CP), hexane extraction (HE), aqueous enzymatic extraction (AEE), and supercritical fluid extraction (SFE). The results indicated that *X. sorbifolia* oil contained a high percentage of monounsaturated fatty acids (49.31–50.38%), especially oleic acid (30.73–30.98%) and nervonic acid (2.73–3.09%) and that the extraction methods had little effect on the composition and content of fatty acids. *X. sorbifolia* oil is an excellent source of nervonic acid. Additionally, the HE method resulted in the highest oil yield (98.04%), oxidation stability index (9.20 h), tocopherol content (530.15 mg/kg) and sterol content (2104.07 mg/kg). The DPPH scavenging activity rates of the oil produced by SFE was the highest. Considering the health and nutritional value of oils, HE is a promising method for *X. sorbifolia* oil processing. According to multiple linear regression analysis, the antioxidant capacity of the oil was negatively correlated with sterol and stearic acid content and positively correlated with linoleic acid, arachidic acid and polyunsaturated fatty acid content. This information is important for improving the nutritional value and industrial production of *X. sorbifolia.*

## 1. Introduction

*Xanthoceras sorbifolia* Bunge is a unique woody oil-rich shrub found in northern China which grows in barren slope valleys and hilly areas at an altitude of 300–2000 m [1], and is known as the “northern *Camellia oleifera*”. The *X. sorbifolia* tree is an eco-friendly tree species with excellent biological characteristics, such as tolerance to drought, temperature, barren land, and mild saline-alkali conditions [2]. Because of the huge geographical span in China, existing studies have found that there are differences between the physical and chemical indicators, fatty acid composition and nutritional elements of *X. sorbifolia* kernels among different accessions [3]. In traditional Chinese medicine, *X. sorbifolia* is used to treat heart diseases, vascular diseases, enuresis, diarrhea, hair loss, skin diseases, mental retardation, Alzheimer’s disease, and other diseases. *X. sorbifolia* kernel oil has been increasingly applied to biodiesel production, supported by the Chinese government, as well as cosmetics, healthy diets, and the food industry. Recently, with the growing awareness of health care, people have started to focus on *X. sorbifolia* oil, which has excellent nutritional value and health benefits because of its high content of unsaturated fatty acids (UFAs), tocopherols and sterols.

Previous studies have found that the oil yield of *X. sorbifolia* kernel is 53.67% and UFAs account for the majority of this [4], especially oleic acid (C_18:1(n−9)_) and linoleic acid (C_18:2(n−6)_). Studies have reported that C_18:1(n−9)_ has beneficial effects on heart diastolic function and plays an important role in metabolism [5]. In addition, C_18:2(n−6)_ is an essential fatty acid for the human body that lowers cholesterol, decreases blood pressure and prevents myocardial infarction [6]. *X. sorbifolia* kernel oil also contains a special fatty acid, nervonic acid (NA, C_24:1_), a very long-chain monounsaturated fatty acid (MUFA) and a major fatty acid in the central nervous system that plays an important role as an antioxidant mediator in human brain cells [7]. As a component of cerebral white matter, NA helps to ensure the proper functioning of nerve fibers and nerve cells. It is difficult for the human body to synthesize NA by itself and it needs to be supplemented in the daily diet. Only a few known plants, such as *Malania oleifera*, *Lunaria annua*, *Borago officinalis*, *Cannabis sativa*, *Tropaeolum speciosum*, *Cardamine graeca*, and *Xanthocera sorbifolium* have been found to contain NA [8]. Hence, *X. sorbifolia oil*, as a high-quality resource, has great value for research.

Current mainstream approaches for oil extraction include hexane extraction (HE), cold-pressing extraction (CP), aqueous enzymatic extraction (AEE), and supercritical fluid extraction (SFE). As an ecologically friendly process, CP is one of the oldest and most commonly used methods for extracting vegetable oils, for which processing conditions include a low temperature and freedom from use of any chemicals or other operations [9]. HE is the most widely used extraction assay to produce commercial vegetable oil owing to its high oil yield [10], low cost, and simple operation. Jahirul et al. [11] compared the extraction processes of CP and HE and found that HE requires an additional refining step and produces extra emissions from volatile organic compounds. Environmental problems, such as solvent loss and associated pollution, as well as safety issues, are the main concerns related to conventional solvent-based oil extraction. These considerations have prompted the development of additional extraction methods. Green biological extraction is currently the mainstream direction in the food industry. AEE is an environmentally friendly method for the simultaneous extraction of oils, proteins, and carbohydrates from cells, without the use of toxic solvents [12]. One disadvantage of AEE is that its product is an oil-in-water emulsion, which requires the addition of a separation step to obtain the oil [13]. In AEE, there is no potential for chemically based oil dissolution, and extraction is based on the insolubility of oil in water rather than the dissolution of oil [14]. Regardless of the supercritical nature and liquid, carbon dioxide extraction is economical and environmentally friendly [15]. Carbon dioxide can be used to replace the organic solvents used in traditional extraction, and is a non-toxic, non-flammable, recyclable, and inexpensive solvent that is easy to remove from the extracts. Moreover, carbon dioxide reaches its critical point under comparatively moderate conditions in the absence of oxygen, which is helpful in maintaining the bioactive compounds [16]. Each of the four methods has advantages and disadvantages. However, there are no reports comparing the effects of different processes on the quality of *X. sorbifolia* oil. In this study, the AEE method, as a new extraction method that had never previously been used to extract *X. sorbifolia* kernel oil, was compared with the CP, HE, and SFE methods.

During processing, *X. sorbifolia* kernel oil is easily oxidized, resulting in the formation of oxygen-containing triglycerides, free fatty acids, and other substances, resulting in a decline in oil quality [17]. Lipid oxidation also produces free radicals and hydroperoxides that can be converted into toxic aldehydes and ketones in the body. The long-term ingestion of these compounds can damage enzymatic systems, resulting in decreased cellular function and tissue necrosis, causing physiological abnormalities and promoting aging, and might ultimately exert carcinogenic effects [18]. 2,2-Diphenyl-1-picrylhydrazyl (DPPH) can be used to measure the free radical scavenging activity of antioxidants, and the oxidative stability index (OSI) reflects the oxidative stability of the oil. These indicators can be used to characterize the antioxidant capacity of *X. sorbifolia* oil from two perspectives, namely shelf life and physiological activity. Therefore, this study comprised a comprehensive comparison and analysis of the basic physical and chemical indicators, fatty acids, minor components, and antioxidant capacity associated with the four methods of *X. sorbifolia* kernel oil extraction. The ultimate aim was to establish a new *X. sorbifolia* kernel oil biological system with nutritional indicators to retain more nutrients and functional properties and to provide theoretical support for the development and application of *X. sorbifolia* kernel oil for large-scale industrial production. A further goal was to predict the main components that affect the antioxidant capacity of the oil through multiple linear regression analysis and to provide key control indicators for optimizing processing technology.

## 2. Materials and Methods

### 2.1. Materials and Chemicals

*X. sorbifolia* (*X. sorbifolia*, 5930240001, Classification and Codes of Chinese Plants) was collected from Jingyuan, Gansu Province. Cellulase and pectinase were purchased from the Pangbo Biological Engineering Co., Ltd. (Nanning, China). 2,2-Diphenyl-1-picrylhydrazyl (DPPH) was purchased from the Lianshuo Biotechnology Co., Ltd. (Shanghai, China). Other reagents and solvents, including n-hexane, fatty acid methyl esters, α-, β-, γ-, and δ- tocopherol and 5α-cholestane standards were purchased from the Sinopharm Chemical Reagent Co., Ltd. (Shanghai, China).

### 2.2. Oil Extraction Methods

*X. sorbifolia* was obtained by hand-shelling, removing impurities, and dried at 60 °C for 3 days in an oven (GZX-9070, Shanghai, China).

HE: n-Hexane was used to extract the oil; 150 g of kernel powder was wrapped in filter paper and placed in the extractor with a ratio of material to n-hexane of 1:5 (*w*/*v*). The extraction was performed at 50 °C for 4 h. After extraction, n-hexane containing the lipid was transferred to a rotary evaporator (RE-52CS, Shanghai, China) until the extraction solvent was completely removed and the oil was collected.

CP: Kernels of 150 g total weight were directly squeezed using a TK-0.5 single screw press (Qufu, Shandong, China) at room temperature with 60 ± 4 °C running temperature and 40 ± 2 °C oil exit temperature. In order to increase friction, 10% shell was added as backfill.

AEE: An aqueous enzymatic extraction method described by Goula et al. (Goula et al., 2018), with some modifications, was used. The experimental conditions were optimized based on previous studies in the laboratory. A quantity of 150 g of kernel powder was commixed with distilled water in a ratio of 1:5 (*w*/*v*) and a DF-101S heat-collecting magnetic stirrer (Shanghai, China) was used to incubate the mixture at 50 °C. The extraction was performed in 3% of cell wall polysaccharide hydrolase (*w*/*w*, cellulase and pectinase ratio 1:2) to effect enzymatic hydrolysis. The amount of enzyme added was 1500 U/g for 3 h with 1% alkaline protease enzymatic hydrolysis (*w*/*w*) for 1 h. After extraction, the mixture was centrifuged (TD5A, Hunan, China) to separate free oil in the bottle from the emulsion phase. The upper layer of free oil was withdrawn using a micropipette, and the emulsion phase was frozen, thawed, centrifuged, and demulsified to obtain the remaining free oil. Then the remaining free oil was added to the bottle.

SFE: A quantity of 150 g of weighed kernel powders was put into an extraction kettle in a HA121-50-01 supercritical carbon dioxide fluid extraction device (Nantong, Jiangsu, China). The supercritical carbon dioxide flow rate was determined to be 18 L/h according to the experimental device conditions. The oil was recovered at an extraction pressure of 28 MPa and a temperature of 42 °C for 192 min in the separation kettle.

### 2.3. Determination of Basic Indicators

Determination of the basic indicators of *X. sorbifolia* followed the Association of Official Analytical Chemists (AOAC) methods. The AOAC standards 948.22, 950.48, 950.49, 925.40, and 935.53 were used to determine the crude fat, crude protein, ash, moisture, and crude fiber content, respectively.

### 2.4. Determination of Physicochemical Properties

The oil yield was recorded based on the weight of oil extracted and the weight of crude fat at room temperature (25 °C). The percentage (wt%) oil yield was obtained gravimetrically using the following equation:$$\mathrm{Oil}\;\mathrm{yield}\;\left(\mathrm{wt}\%\right)=\frac{\mathrm{weight}\;\mathrm{of}\;\mathrm{oil}\;\mathrm{extracted}\;\left(g\right)}{\mathrm{weight}\;\mathrm{of}\;\mathrm{crude}\;\mathrm{fat}\;\left(g\right)}$$

The moisture and volatile matter, acid value (AV) and peroxide value (PV) were determined in accordance with AOAC standards 969.17, 965.33 and 926.12, respectively. The OSI was determined in accordance with AOCS standard Cd 12b-92.

### 2.5. Analysis of Fatty Acid Composition

An Agilent 7890A gas chromatograph (Shanghai, China) and an SP-2560 column (100 m × 0.25 mm × 0.20 μm, Supelco, Bellefonte, PA, USA) were used to analysis the fatty acid composition. The preparation for fatty acid methylation and the analysis conditions for the oil samples were as described by Gao et al. [17] with some modifications. A quantity of 50 mg of oil was dissolved in 2 mL petroleum ether and mixed with 2 mL of 0.4 mol L^−1^ potassium hydroxide methanol solution to derivatize fatty acids into methyl esters. The detailed specifications for GC analysis were as follows: injection volume was 1.0 μL; carrier gas was high purity helium; split ratio was 1:50; inlet temperature was 250 °C; detector temperature was 280 °C; hydrogen flow rate 40 mL/min; air flow rate 400 mL/min; auxiliary gas flow rate 40 mL/min; the temperature rising program was: 100 °C for 5 min, increased to 175 °C at 10 °C/min, kept for 10 min, then increased to 230 °C at 5 °C/min, kept for 20 min. The retention times of 37-standard peak were used as an indicator for qualitative analysis and measurement of the peak area was used as an indicator for quantitative analysis.

### 2.6. Determination of Nutritional Components

#### 2.6.1. Tocopherol Content

The method previously reported by Gao et al. [17], with some improvements, was used to determine tocopherol content. Tocopherols were analyzed using a high-performance liquid chromatographic system (Agilent 1200HPLC, Santa Clara, CA, USA) with the UV excitation and excitation wavelength at 298 nm and the emission wavelength at 330 nm. The separation was performed on a silica column (5 μm, 4.6 × 250 mm, Hanbon, Jiangsu, China) using hexane/isopropanol (98/2, *v*/*v*) as the mobile phase, with a rate of 0.8 mL/min for 30 min. α-, β-, γ-, and δ-tocopherols were identified and quantified by reference to standards, and their contents were reported in mg/kg of oil.

#### 2.6.2. Sterol Content

Sterol was analyzed using a gas chromatograph-mass spectrum system (7890B, Agilent, Santa Clara, CA, USA). First, 0.25 g of oil was mixed with 1 mL 0.1 mg/mL 5α-cholestane standard solution and 5 mL of 0.5 mol/L KOH-CH_3_CH_2_OH (potassium hydroxide dissolved in ethanol). The mixture was heated connecting with reflux condensation for 15 min and then 5 mL of ethanol was immediately added. A quantity of 5 mL of the above solution was pipetted into the prepared alumina column. The unsaponifiable matter was first eluted with 5 mL of ethanol and then eluted with 30 mL of ether. The flow rate was approximately 2 mL/min. A quantity of 1 μm of the sample was injected into the gas chromatograph, separated by a SE-54 capillary column (0.25 μm, 50 m× 0.25 mm, Agilent, Shanghai, China). The carrier gas flow rate was 36 cm/s and the split ratio was 1:20. The initial column temperature was set at 240 °C and then increased to 255 °C at a rate of 4 °C/min. The sterol content was reported in mg/kg of oil.

### 2.7. Determination of DPPH Free Radical Scavenging Rate

DPPH assay assesses the ability to reduce a stable radical through determination of decrease in absorbance. A DPPH assay was conducted according to the method of Shi et al. [19] with some modifications. A quantity of 20 mg of DPPH was accurately weighed, diluted in a volumetric flask with 500 mL of absolute ethanol, shaken well and stored in a dark place at room temperature. Then quantities of the oil samples were weighed to prepare test solutions with different mass concentrations. A quantity of 0.2 mL of the test solution and 3.8 mL DPPH was pipetted out accurately, shaken and mixed well, then reacted for 1 h in the dark. Another 0.2 mL of water was mixed with 3.8 mL of DPPH and the absorbance A_0_ was determined; the absorbance A_r_ was measured for 0.2 mL of the test solution mixed with 3.8 mL of absolute ethanol. The calculation formula used for determination of the DPPH radical scavenging rate was as follows:$$\mathrm{DPPH}\;\mathrm{radical}\;\mathrm{scavenging}\;\mathrm{rate}=(1-\frac{A_{s}\;-\;A_{r}}{A_{0}})\;\times \;100\%$$

A_s_: absorbance of test solution and DPPH measured at 517 nm;A_r_: absorbance of test solution and absolute ethanol measured at 517 nm;A_0_: absorbance of ethanol and DPPH measured at 517 nm.

### 2.8. Statistical Analysis

All the experimental tests were performed in triplicate, and the results were reported as means ± standard deviations. Statistical analysis was performed using SPSS 19.0 (IBM, Armonk, NY, USA). One-way analysis of variance (ANOVA) was used for data comparison, and the difference between variables was assessed using Duncan’s multiple range test (*p* < 0.05). Multiple linear regression analysis was used to determine the correlations between antioxidant capacity assays and minor components using a stepwise method. All variables were standardized by transformation of statistical values into z-scores before the employment of chemometrics.

## 3. Results and Discussion

### 3.1. Chemical Composition

The basic chemical composition of the *X. sorbifolia* seeds is shown in Table 1. The crude fat content of kernels was 58.16%, which was higher than that of most woody plants, such as *Camellia* seed (38.39%) [20], Amygdalus pedunculata Pall kernel (50.57%) [8] and Acer truncatum Bunge (47.60%) [21]. The kernels also had a higher crude protein content (29.53%) than palm kernel (19.16%), peanut (22.18%), and Acer truncatum Bunge (27.15%). Therefore, *X. sorbifolia* is a potentially valuable resource for further development.

### 3.2. Physicochemical Properties

Table 2 shows the oil yield, moisture and volatile matter, AV, PV, and OSI of *X. sorbifolia* kernel oil. The oil yield with different production methods varied greatly; HE provided the highest yield of oil compared to that obtained with other extraction approaches, with a yield of 98.04%—the approach with the lowest yield was AEE (68.74%). The oil extraction rate of AEE was much lower than that of the other methods, and this trend was similar to that observed with other crops [14,22]. Lipid yields decreased in the following order: HE > SFE > CP > AEE. The moisture and volatile matter content of *X. sorbifolia* oil produced by AEE were extremely high (1.81%). As the enzymatic hydrolysis system of AEE comprises a water environment, further refinement is required. As shown in Table 2, it was found that the AV of *X. sorbifolia* kernel oil ranged from 0.12 to 0.58 mg KOH/g, whereas the peroxide value, which was much lower than the value for commercial edible vegetable oil (≤10 mmol/kg), conformed to the standard of edible vegetable oil. A low acid value indicated that the proportion of free fatty acids among the total lipids was low. SFE resulted in the highest AV (0.58 mg/g), which might be due to the longer high-pressure treatment time with SFE in the supercritical carbon dioxide environment, and the free fatty acid content increased. The physical and chemical properties of *X. sorbifolia* kernel oils processed using different methods were different, among which the CP and AEE oils exhibited the best properties. The OSI was defined as the point of maximal change in the oxidation rate, which was attributed to the increase in conductivity owing to the formation of volatile organic acids during lipid oxidation [23]. The oil prepared by HE had the highest OSI value (9.2 h), which was significantly higher than that of the other three extraction processes. The AEE oil (8.53 h) ranked second, whereas CP (7.85 h) and SFE (7.86 h) oils had lower values, but with no significant difference (*p* < 0.05). Other studies have indicated that oil obtained from SFE is markedly less stable than that obtained by HE [24]. Several factors contribute to the oxidative stability of fats and oils. Previous investigations have shown that SFE removes some substances that could act as metal chelators, which are precursor materials of Maillard reaction products, or as synergists that function with natural antioxidants during processing [25], and this might be the cause of the observed low oxidative stability.

### 3.3. Fatty Acid Composition

The fatty acid composition of the oil samples prepared by different methods are shown in Table 3. The composition of fatty acids prepared using different methods was the same, but the percentage content was slightly different. *X. sorbifolia* kernel oil was extremely rich in UFAs, including MUFAs (C_18:1(n−9)_: 30.73–30.98%) and polyunsaturated fatty acids (PUFAs, C_18:2(n−6)_: 40.55–40.86%). The MUFA content obtained by AEE was the highest (50.38%), followed by that with SFE (49.81%). The PUFA content obtained by the HE method was the highest (41.78%), followed by that obtained with the AEE method (41.69%). This result is consistent with that of previous studies [26,27]. In a previous study [21] on *X. sorbifolia*, provenance had a significant influence on the fatty acid composition and content of *X. sorbifolia* kernel oil. It was concluded that the difference in origin was a significant factor affecting the fatty acid composition and content of the oil, regardless of the extraction method.

The NA content in *X. sorbifolia* kernel oil accounted for 3% of the total, similar to that reported in a previous study [1]. NA is a fatty acid with high concentrations found in nerve and brain tissues. It can restore nerve ending activity, promote the growth and development of nerve cells, and is essential for the growth, development, and repair of the cerebral nervous system [28]. NA-containing oils are important for the prevention and treatment of neurological disorders. Researchers have found that the NA content in vegetable oil is very low, and that only a few plants contain NA. Three unique NA-containing plants have been discovered in China, namely *M. oleifera*, *A.*
*truncatum* Bunge and *X. sorbifolia*. The oil content in *M. oleifera* was up to 60%, with an NA content of approximately 50%. Regretfully, this species is currently endangered and only exists in parts of Yunnan and Guangxi provinces in China. The content of NA in *A. truncatum* Bunge oil was 5% [29], but it has strict requirements for its growth environment. Large tracts of desert in China have not been properly developed and *X. sorbifolia* possesses characteristics of drought and wind resistance. Therefore, *X. sorbifolia* kernel oil has high development value and is a potential new source of NA.

### 3.4. Nutrient Content

The tocopherol and sterol contents of *X. sorbifolia* kernel oil prepared by different methods are shown in Table 4. γ-tocopherol was the major tocopherol found in the oil (329.18–361.37 mg/kg), accounting for more than 68% of total tocopherol. The α-, γ-, δ-tocopherol and total tocopherol contents of the oil obtained by HE was the highest (94.51, 361.37, 74.27, and 530.15 mg/kg, respectively). Tocopherol is temperature sensitive and has poor water solubility and stability. Decrease in tocopherol is mainly caused by high temperature and the length of time required for the extraction process. Hence, in the AEE process, total tocopherol might be degraded because of the influence of pH, temperature and the time taken for enzymatic hydrolysis, while the CP method is not conducive to the dissolution of tocopherol. Moreover, as the biologically active form and the major liposoluble antioxidant in membrane structures, α-tocopherol was found in high content in *X. sorbifolia* kernel oil. The sterol content in the *X. sorbifolia* kernel oil obtained by HE was the highest (2105.06 mg/kg), followed by that obtained with SFE (2004.51mg/kg), whereas the observed contents with AEE and CP were relatively lower (1276.55 and 1340.20 mg/kg, respectively).

It was found that the content of tocopherols and sterols in *X. sorbifolia* kernel oil was higher than that of oil crops, such as walnut and linseed, and was similar to that of *Camellia oleifera* seed oil [30]. In summary, HE resulted in the best physiological effects, followed by SFE. Modern medical research has confirmed that tocopherol has anti-oxidation and anti-cancer effects and prevents myopia, while sterols can lower cholesterol and inhibit cancer and cardiovascular diseases. Therefore, *X. sorbifolia* oil is a good dietary nutrient source.

### 3.5. DPPH Scavenging Activity

The DPPH radical scavenging assay is most frequently used to evaluate antioxidant activity. The DPPH assay is considered to be mainly based on an electron transfer reaction, and hydrogen atom abstraction is a marginal reaction pathway [31]. With increase in the oil sample concentration, the scavenging ability of the four types of *X. sorbifolia* kernel oil for DPPH free radicals gradually increased. As shown in Figure 1, the DPPH free-radical-scavenging ability of the oil prepared by SFE was relatively strong, from 39.27% (sample concentration at 1 mg/mL) to 67.50% (sample concentration at 10 mg/mL). Compared to other oils, the strongest free-radical-scavenging ability observed with SFE was 3.37%, 7.44%, and 10.06% higher than that with AEE, CP, and HE, respectively.

It is noteworthy that the HE oil had the weakest antioxidant capacity, while it had the highest lipid yield, which may indicate that the HE method was accompanied by the loss of parts of antioxidant substances while obtaining a high lipid yield.

### 3.6. Multiple Linear Regression Analysis

Since *X. sorbifolia* kernel oil exhibited antioxidant capacity and studies have found that antioxidant capacity is related to certain compounds [32], it was expected that one or several chemical components would have a certain correlation with or mutual restriction effect on the antioxidant capacity of the oil. To explore this, we used multiple linear regression analysis and built regression models to identify the endogenous compounds that affect the antioxidant capacity of the oil.

The prediction of the relationship between antioxidant capacity and minor components using multiple linear regression analysis is shown in Table 5. The results showed that DPPH was significantly positively correlated with sterol (R = 0.423; *p* < 0.05) and significantly negatively correlated with C_20:0_ (R = −0.957; *p* < 0.05) and PUFA (R = −0.336; *p* < 0.05); OSI was significantly positively correlated with sterol (R = 0.784; *p* < 0.05) and C_18:2(n−6)_ (R = −0.074; *p* < 0.05) and significantly negatively correlated with C_18:0_ (R = −0.513; *p* < 0.05).

Previous studies have found a relationship between antioxidant capacity and the presence of sterols. Sterols have a structure that allows them to react rapidly with lipid free radicals to form relatively stable free radicals, which are effective as antioxidants. Relatively stable free radicals interrupt the triglyceride autoxidation chain reaction. In the oxidation process of common unsaturated natural vegetable oils and synthetic monoesters, studies have found that the oxidation of unsaturated oils involves an epoxidation reaction, and oxidation is traditionally believed to occur at the double bonds. As the oxidation depth increases, isomerization of the unsaturated structure intensifies; that is, the unsaturated double bond changes from a homeopathic structure to a trans structure. The more C_18:2(n−6)_ and linolenic acid (C_18:3(n−3)_) that are present in the fatty acid of the oil, the greater the degree of isomerization and the faster the oxidation rate [33].

The OSI of vegetable oils depends on their fatty acid composition and the presence of antioxidants, mainly sterols. It is well known that lipid oxidation comprises the extraction of a hydrogen atom from an allylic orbis-allylic position of UFAs to produce alkyl free radicals, which combine with molecular oxygen at a diffusion-controlled rate to produce lipid peroxide free radicals, whereas double bonds act as free radical receptors. The effect of fatty acids on oxidation stability is mainly determined by their unsaturation and, to a lesser extent, the position of unsaturated executive groups in triacylglycerol molecules. Studies have shown that the antioxidant properties of purified oil at 60 °C are positively correlated with the average number of double bonds and the content of C_18:1(n−9)_. Multiple linear regression analysis cannot reveal the synergy or antagonism between the variables and there might be other polar compounds that have not been detected involved in the antioxidant activity, or there might be synergy between the minor components.

## 4. Conclusions

The effects of different extraction processes on the physicochemical properties, nutritional components, and antioxidant ability of *X. sorbifolia* kernel oil were investigated. The oil yield obtained with the HE method was the highest (98.04%). *X. sorbifolia* kernel oil is a good resource for UFAs and nutrients. The oil extraction method did not change the fatty acid composition but did change the percentage content. NA is a special fatty acid contained in oil, and its content in the samples was 2.73–3.09%. *X. sorbifolia* kernel oil is one of the few edible oils available that can provide NA. The total tocopherol and sterol content using the HE method was the highest (530.42 mg/kg and 2105.06 mg/kg, respectively). The scavenging rate of DPPH free radicals from the oil obtained using the SFE method was the highest. According to multiple linear regression analysis, the antioxidant capacity of the oil was mainly related to C_20:0_, PUFAs and sterol, and the OSI of the oil was mainly related to C_18:0_, C_18:2(n−6)_, and sterol. This demonstrates that *X. sorbifolia* is a potential oil resource with rich nutrient content and strong antioxidant capacity. To enable the large-scale utilization of *X. sorbifolia* kernel oil and to realize its industrial production, further evaluation of the color, smell, polyphenol content, and other indicators of the oil is necessary. The selection of processing methods yielding high sterol and C_18:2(n−6)_ contents is an important way to improve the antioxidant capacity of *X. sorbifolia* kernel oil. At present, the identification of antioxidant substances in *X. sorbifolia* kernel oil is relatively straightforward; however, the relationship between tocopherols, sterols and phenolic substances and antioxidant capacity requires further analysis in the future.

## Figures and Tables

**Figure 1 molecules-27-04185-f001:**
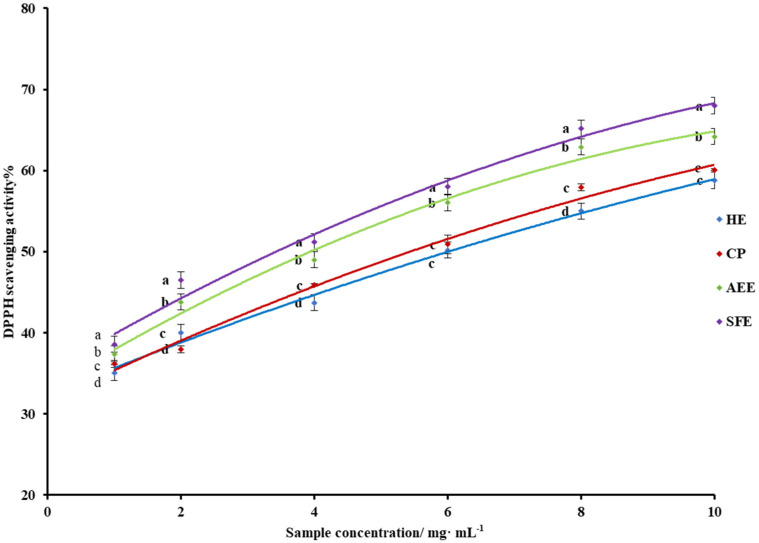
The DPPH radical-scavenging ability assay of *X. sorbifolia* oil. Means with different letters in the same column are significantly different.

**Table 1 molecules-27-04185-t001:** The chemical composition of the *X. sorbifolia* seed.

Composition	Crude Fat	Crude Protein	Moisture	Ash	Crude Fiber
content (%)	58.16 ± 0.18	29.53 ± 0.25	4.43 ± 0.13	2.32 ± 0.09	1.61 ± 0.12

**Table 2 molecules-27-04185-t002:** The physicochemical properties of *X. sorbifolia* oil.

	HE	CP	AEE	SFE
oil yield (%)	98.04 ± 0.29 ^a^	87.81 ± 0.29 ^c^	68.74 ± 0.18 ^d^	89.63 ± 0.17 ^b^
moisture and volatile matter (%)	0.02 ± 0.00 ^b^	0.02 ± 0.00 ^b^	1.81 ± 0.05 ^a^	0.01 ± 0.00 ^b^
AV (KOH)/(mg/g)	0.19 ± 0.02 ^c^	0.12 ± 0.01 ^d^	0.25 ± 0.02 ^b^	0.58 ± 0.02 ^a^
PV (mmol/kg)	0.58 ± 0.02 ^b^	0.65 ± 0.03 ^a^	0.47 ± 0.02 ^d^	0.51 ± 0.01 ^c^
OSI (h)	9.20 ± 0.05 ^a^	7.85 ± 0.04 ^c^	8.53 ± 0.18 ^b^	7.86 ± 0.06 ^c^

Different letters in the same row indicate significant statistical differences. The same letters indicate no significant difference (Tukey’s test, *p* < 0.05).

**Table 3 molecules-27-04185-t003:** The fatty acid composition (%) of *X. sorbifolia* oil.

	HE	CP	AEE	SFE
C_16:0_	5.40 ± 0.061 ^a^	5.23 ± 0.087 ^b^	4.60 ± 0.061 ^c^	5.31 ± 0.064 ^a,b^
C_18:0_	2.22 ± 0.031 ^c^	2.26 ± 0.070 ^b^	2.12 ± 0.031 ^d^	2.32 ± 0.061 ^a^
C_18:1(n−9)_	31.48 ± 0.055 ^a,b^	31.36 ± 0.065 ^b,c^	31.28 ± 0.090 ^c^	31.53 ± 0.046 ^a^
C_18:2(n−6)_	40.55 ± 0.036 ^b^	40.58 ± 0.060 ^b^	40.86 ± 0.101 ^a^	40.59 ± 0.089 ^b^
C_18:3(n−3)_	0. 50 ± 0.017 ^a^	0.46 ± 0.058 ^b^	0.40 ± 0.026 ^b^	0.40 ± 0.026 ^b^
C_20:0_	0.44 ± 0.010 ^a^	0.38 ± 0.025 ^b^	0.28 ± 0.029 ^d^	0.32 ± 0.029 ^c^
C_20:1_	6.88 ± 0.061 ^b^	6.89 ± 0.040 ^b^	7.13 ± 0.078 ^a^	6.84 ± 0.067 ^c^
C_20:2_	0.38 ± 0.021 ^b^	0.43 ± 0.030 ^a^	0.43 ± 0.030 ^a^	0.37 ± 0.023 ^c^
C_22:0_	0.54 ± 0.015 ^c^	0.58 ± 0.021 ^a,b^	0.59 ± 0.010 ^a^	0.57 ± 0.023 ^b^
C_22:1_	8.22 ± 0.026 ^c^	8.47 ± 0.032 ^b^	8.88 ± 0.081 ^a^	8.44 ± 0.078 ^b^
C_24:0_	0.31 ± 0.006 ^b^	0.34 ± 0.052 ^a^	0.34 ± 0.052 ^a^	0.31 ± 0.006 ^b^
C_24:1_	2.73 ± 0.025 ^c^	3.02 ± 0.075 ^b^	3.09 ± 0.047 ^a^	3.03 ± 0.085 ^b^
SFA	8.91 ± 0.07 ^a^	8.77 ± 0.07 ^b^	7.93 ± 0.07 ^c^	8.83 ± 0.05 ^a,b^
MUFA	49.31 ± 0.05 ^c^	49.75 ± 0.11 ^b^	50.38 ± 0.17 ^a^	49.81 ± 0.04 ^b^
PUFA	41.78 ± 0.03 ^a^	41.48 ± 0.04 ^b^	41.69 ± 0.11 ^a^	41.37 ± 0.08 ^b^

Different letters in the same row indicate significant statistical differences. The same letters indicate no significant difference (Tukey’s test, *p* < 0.05). SFA (C_16:0_ + C_18:0_ + C_20:0_ + C_22:0_ + C_24:0_), MUFA (C_18:1_ + C_20:1_ + C_22:1_ + C_24:1_), PUFA (C_18:2__(n−6)_ + C_18:3__(n−3)_ + C_20:2_).

**Table 4 molecules-27-04185-t004:** The tocopherol and sterol content (mg/kg) of *X. sorbifolia* oil.

	HE	CP	AEE	SFE
α-Tocopherol	94.51 ± 0.70 ^a^	73.53 ± 2.18 ^b^	74.28 ± 2.72 ^b^	75.83 ± 4.40 ^b^
γ-Tocopherol	361.37 ± 3.85 ^a^	329.18 ± 7.31 ^b^	333.84 ± 16.22 ^b^	343.69 ± 16.37 ^a,b^
δ-Tocopherol	74.27 ± 0.60 ^a^	49.09 ± 5.09 ^b^	53.78 ± 2.12 ^b^	53.33 ± 2.07 ^b^
Total tocopherol	530.15 ± 5.14 ^a^	451.80 ± 0.74 ^b^	461.90 ± 15.61 ^b^	472.85 ± 18.70 ^b^
Sterol	2104.07 ± 14.44 ^a^	1340.20 ± 31.00 ^c^	1274.71 ± 6.65 ^d^	2004.51 ± 8.93 ^b^

Different letters in the same row indicate significant statistical differences. The same letters indicate no significant difference (Tukey’s test, *p* < 0.05).

**Table 5 molecules-27-04185-t005:** Prediction of the relationship between antioxidant capacity and minor components by multiple linear regression analysis.

Dependent Variable	Variable	R	VIF	Equation
DPPH	(Constant)	−6.548 × 10^−15^		Y = −6.548 × 10^−15^ − 0.957 (C_20:0_) − 0.336 (PUFA) + 0.432 (sterol)
	C_20:0_	−0.957	1.576	
	PUFA	−0.336	1.164	
	sterol	0.432	1.405	
OSI	(Constant)	1.931 × 10^−16^		Y = 1.931 × 10^−16^ − 0.513 (C_18:0_) + 0.784 (sterol) + 0.074 (C_18:2(n−6)_)
	C_18:0_	−0.513	1.162	
	C_18:2(n−6)_	0.074	1.409	
	Sterol	0.784	1.232	

## Data Availability

Not applicable.

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
