# Peer review of "Comparison of Different Extraction Processes on the Physicochemical Properties, Nutritional Components and Antioxidant Ability of *Xanthoceras sorbifolia* Bunge Kernel Oil"

_molecules, 2022, doi:10.3390/molecules27134185_

Round 1
Reviewer 1 Report
In the submitted manuscript Authors investigated the oil content, physicochemical properties, 15 fatty acid composition, nutrient content, and antioxidant ability of Xanthoceras sorbifolia Bunge (X. 16 sorbifolia) kernel oils obtained by different extraction methods. Generally, the paper is well-constructed and presented clearly, however there are some points that must be clarified before the article can be accepted:
1. Some published recently papers are missing. Please add them to the references list and please have a comparative discussion:
ü Shen, Z., Zhang, K., Ao, Y. et al. Evaluation of biodiesel from Xanthoceras sorbifolia Bunge seed kernel oil from 13 areas in China. J. For. Res. 30, 869–877 (2019). https://doi.org/10.1007/s11676-018-0683-9
ü Ji Li, Yuan-Gang Zu, Meng Luo, Cheng-Bo Gu, Chun-Jian Zhao, Thomas Efferth, Yu-Jie Fu, Aqueous enzymatic process assisted by microwave extraction of oil from yellow horn (Xanthoceras sorbifolia Bunge.) seed kernels and its quality evaluation, Food Chemistry, Volume 138, Issue 4, 2013, Pages 2152-2158, https://doi.org/10.1016/j.foodchem.2012.12.011.
ü Xue-Jin Qu, Yu-Jie Fu, Meng Luo, Chun-Jian Zhao, Yuan-Gang Zu, Chun-Ying Li, Wei Wang, Ji Li, Zuo-Fu Wei, Acidic pH based microwave-assisted aqueous extraction of seed oil from yellow horn (Xanthoceras sorbifolia Bunge.), Industrial Crops and Products, Volume 43, 2013, Pages 420-426, https://doi.org/10.1016/j.indcrop.2012.07.055.
2. What kind of the detector was used to identify the fatty acids components?
3. Please add into the supplementary materials the chromatograms of GC analysis considering fatty acid compositions depending on the used extraction method.
